# The iron–sulphur cluster in human DNA2 is required for all biochemical activities of DNA2

Laura Mariotti [1]✉, Sebastian Wild[1,4], Giulia Brunoldi[1,4], Alessandra Piceni[1,4], Ilaria Ceppi[2,3], Sandra Kummer[1], Richard E. Lutz[1], Petr Cejka[2,3] & Kerstin Gari [1]✉

The nuclease/helicase DNA2 plays important roles in DNA replication, repair and processing of stalled replication forks. DNA2 contains an iron-sulphur (FeS) cluster, conserved in eukaryotes and in a related bacterial nuclease. FeS clusters in DNA maintenance proteins are required for structural integrity and/or act as redox-sensors. Here, we demonstrate that loss of the FeS cluster affects binding of human DNA2 to specific DNA substrates, likely through a conformational change that distorts the central DNA binding tunnel. Moreover, we show that the FeS cluster is required for DNA2's nuclease, helicase and ATPase activities. Our data also establish that oxidation of DNA2 impairs DNA binding in vitro, an effect that is reversible upon reduction. Unexpectedly, though, this redox-regulation is independent of the presence of the FeS cluster. Together, our study establishes an important structural role for the FeS cluster in human DNA2 and discovers a redox-regulatory mechanism to control DNA binding.

[1] Institute of Molecular Cancer Research, University of Zurich, 8057 Zurich, Switzerland. [2] Institute for Research in Biomedicine, Faculty of Biomedical Sciences, Università della Svizzera italiana, 6500 Bellinzona, Switzerland. [3] Department of Biology, Institute of Biochemistry, ETH Zurich, 8092 Zurich, Switzerland. [4] These authors contributed equally: Sebastian Wild, Giulia Brunoldi, Alessandra Piceni. ✉email: mariotti@imcr.uzh.ch; gari@imcr.uzh.ch

Iron–sulphur (FeS) clusters are ancient inorganic protein cofactors best known for their role in electron transport in the mitochondrial respiratory chain[1]. The maturation of FeS clusters and their incorporation into proteins is a multi-step process that relies on mitochondria and, in the case of cytosolic and nuclear FeS proteins, on the cytoplasmic iron–sulphur assembly (CIA) machinery[2–5]. Due to their redox reactivity FeS clusters are intrinsically vulnerable to oxidation, which may release iron ions that can generate reactive oxygen species[4]. It is therefore surprising to find them in many proteins involved in DNA replication and repair, including the DNA helicases DDX11[6], XPD and FANCJ[7], DNA primase[8,9], DNA glycosylases[10,11], and DNA polymerases alpha, delta and epsilon[12]. In some DNA maintenance proteins, FeS clusters act as important structural components involved in protein complex formation[12] or supporting catalytic and non-catalytic activities of their host proteins[7,8,13]. In others, they seem to have an active redox role, such as in the bacterial DNA damage-inducible protein DinG, where reduction of its FeS cluster reversibly inactivates its helicase activity[14]. Although the precise roles of FeS clusters often remain elusive, their abundance in DNA maintenance proteins and the fact that they have not been replaced with less redox-sensitive metals throughout evolution suggest they play an important role in DNA processing enzymes.

The presence of a 4Fe-4S cluster in the nuclease/helicase DNA2 was initially suggested due to its homology to the bacterial AddB nuclease, which dimerises with AddA[15]. DNA2 is a single-stranded DNA (ssDNA) endonuclease and helicase with roles in DNA replication[16–18], DNA double-strand break (DSB) repair[19–21] and restart of stalled replication forks[22,23]. In the related AddAB, it was discovered that loss of the FeS cluster affects binding to double-stranded DNA (dsDNA) and its nuclease and dsDNA-dependent ATP activities[15]. The coordination of the FeS cluster in AddB and DNA2 is unusual, with one FeS cluster-binding cysteine located a few hundred amino acids upstream of the other three[15]. Since the purified FeS cluster-deficient AddAB variants were less stable than their wild-type counterpart, it was suggested that the FeS cluster acts as a staple that pins back the nuclease domain to support its structural integrity[15]. Similar to AddAB, FeS cluster-deficient yeast Dna2 had reduced nuclease and ATPase activities, whereas DNA binding and protein stability did not seem impaired[24]. The crystal structure of mouse Dna2 confirmed that the FeS cluster supports a crossover loop in the nuclease domain, which forms the base of a cylinder into which ssDNA is threaded[25]. It was also shown that replication protein A (RPA), which recruits Dna2 to 5′-tailed ssDNA[26,27], binds Dna2 at a flexible N-terminal helix and the OB domain, which are both located on the outside of the cylinder[25]. It is therefore not surprising that loss of the FeS cluster in yeast Dna2 did not affect binding to RPA[24].

In this study, we demonstrate that loss of the FeS cluster in human DNA2 induces a conformational change that likely distorts the DNA-binding tunnel and consequently impairs DNA binding. Nuclease, helicase and ATPase activities are also affected by FeS cluster loss. Finally, we discover that DNA2 is redox-regulated, but, in contrast to the helicase DinG[14], this redox sensitivity is not mediated by its FeS cluster. This surprising finding adds DNA2 to the growing list of nuclear proteins which may act as redox sensors.

## Results

**FeS cluster binding is not required for overall stability.** To obtain FeS cluster-deficient versions of human DNA2, we replaced each of the four ligating cysteines with serine residues (Fig. 1a). We also generated double-cysteine and triple-cysteine

variants, since some FeS cluster-containing proteins with single cysteine substitutions can still incorporate iron[8]. We assessed the iron content of the purified proteins using a radioactive iron incorporation assay (Fig. 1b and Supplementary Fig. 1a). Only wild-type DNA2 incorporated iron-55 above background levels, suggesting that the six cysteine variants are all FeS

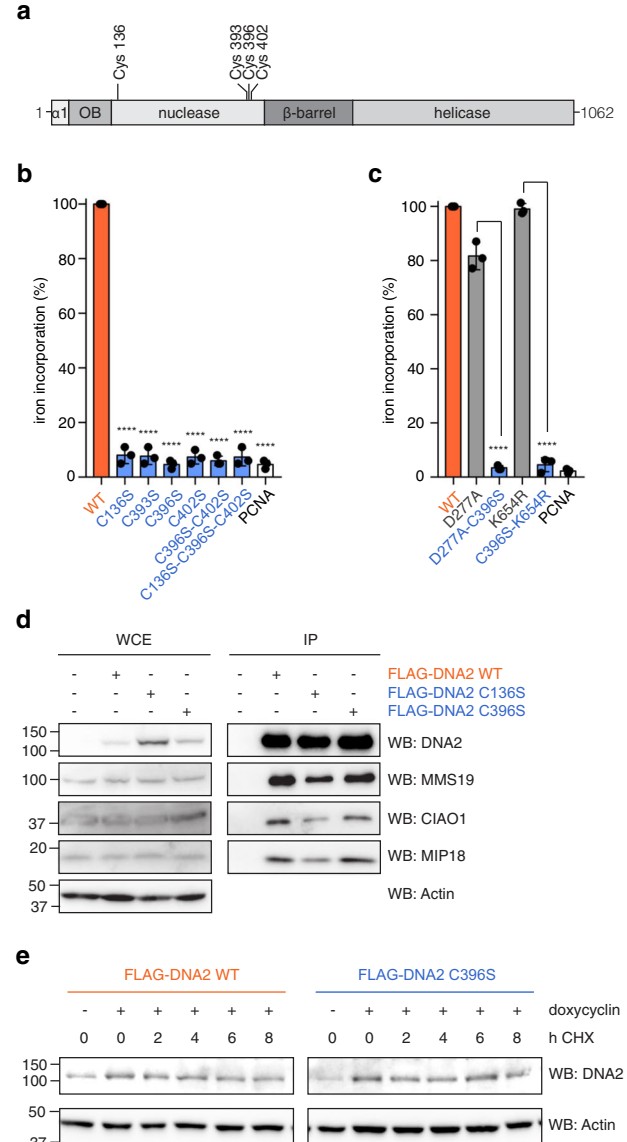

**Fig. 1 Characterisation of the FeS cluster-deficient DNA2 variants.**
**a** Domain organisation of human DNA2, with the four FeS cluster-coordinating cysteines highlighted. **b** Radioactive iron-55 incorporation in wild-type DNA2 and cysteine variants, measured by liquid scintillation counting. Data are expressed relative to wild-type, which is set to 100% ($n = 3$ independent experiments; error bars, SD). Statistical analysis: ordinary one-way ANOVA (****$p < 0.0001$). **c** Iron incorporation as in **b** for wild-type, nuclease-deficient (D277A) and helicase-deficient (K654R) DNA2, with or without FeS cluster-deficiency (C396S) ($n = 3$ independent experiments; error bars, SD). Statistical analysis: ordinary one-way ANOVA (****$p < 0.0001$). **d** FLAG-DNA2 WT, C136S and C396S over-expressed in HEK293T cells were immunoprecipitated, and co-immunoprecipitation with endogenous MMS19, CIAO1 and MIP18 was assessed by SDS–PAGE and Western blotting. **e** Stability of inducibly expressed FLAG-DNA2 WT and FLAG-DNA2 C396S over time upon addition of cycloheximide (CHX), analysed by SDS–PAGE and Western blotting.

cluster-deficient. We also confirmed that nuclease-deficient (D277A) and helicase-deficient (K654R) variants of DNA2, used in this and previous studies[25,28–30], had intact FeS clusters (Fig. 1c and Supplementary Fig. 1b).

It was shown by SDS–PAGE analysis that loss of the FeS cluster in the bacterial helicase DinG results in protein degradation[14]. The purified DNA2 proteins, including the FeS cluster-deficient variants, appeared however stable in SDS–PAGE analyses (Supplementary Fig. 1c). The mutations did not seem to impair folding in a cellular context either, since FLAG-DNA2 C136S and C396S over-expressed in HEK293T cells were able to interact with the CIA-targeting complex (responsible for loading the FeS cluster onto apo client proteins) in pull-down experiments (Fig. 1d). Finally, to compare the stability of wild-type and FeS cluster-deficient DNA2 over time, expression of FLAG-DNA2 WT and C396S was induced by doxycycline treatment, and protein translation was blocked by addition of the translation inhibitor cycloheximide 16 h after induction. Levels of FLAG-DNA2 WT and C396S were similar up to 8 h after cycloheximide treatment (Fig. 1e), indicating that in this experimental setup DNA2's protein stability is not compromised by loss of the FeS cluster.

**Reduced DNA binding through a conformational change**. We then wondered whether loss of the FeS cluster affects DNA binding. The crystal structure of mouse Dna2 revealed that the DNA-binding tunnel can only accommodate ssDNA[25] and the affinity of human and yeast DNA2/Dna2 for various DNA substrates has been well-characterised[16,24,25,29]. In line with these previous studies, wild-type DNA2 readily shifted a 5′ flap oligonucleotide-based DNA substrate in electrophoretic mobility shift assays (EMSAs) at relatively low protein concentrations (Fig. 2a). In contrast, a 10-fold higher protein concentration was required for DNA2 C396S to achieve comparable DNA binding (Fig. 2a). A similar reduction in DNA binding was also observed with the other cysteine variants (Fig. 2b). It is of note that in our assay conditions a considerable portion of DNA2/DNA complexes did not enter the polyacrylamide gel at protein concentrations higher than 50 nM (Fig. 2a,b). This suggests that the complexes may form higher-order assemblies, an effect already seen for DNA2 elsewhere[29].

The ~10-fold reduction in DNA binding that we observed with the FeS cluster-deficient DNA2 variants may indicate that the affinity for DNA is greatly reduced in the absence of an FeS cluster and that DNA binding is only favoured at high protein concentrations. However, to rule out the possibility that only a subpopulation of the FeS cluster-deficient DNA2 variants is properly folded and, hence, proficient in DNA binding, whereas the majority is misfolded and tends to aggregate, we analysed both wild-type DNA2 and the C396S variant by size exclusion chromatography (Supplementary Fig. 2a). Although in both cases some protein was found in the void peak, which is indicative of protein aggregation, the majority of wild-type DNA2 and the C396S variant eluted as a single peak after the void fractions. Surprisingly, however, the C396S variant eluted at a higher estimated molecular weight from the gel filtration column than the wild-type protein.

To explain this observation, we considered the following possibilities: (1) an FeS cluster loss-induced oligomerisation of the C396S variant or (2) a conformational change upon FeS cluster loss. We could exclude an altered homo-oligomerisation behaviour upon FeS cluster loss since neither YFP-DNA2 (WT) nor YFP-DNA2 C396S was able to pull down purified DNA2 (WT/C396S) (Supplementary Fig. 2b). We therefore presume that the different elution patterns in size exclusion chromatography

most likely result from a conformational change upon FeS cluster loss. Given that the FeS cluster in DNA2 is coordinated by distant cysteines and known to pin back the nuclease domain[15,25], a major conformational change that leads to a more elongated shape seems plausible.

In line with a conformational change upon FeS cluster loss, the protein/DNA complexes formed by wild-type DNA2 and the DNA2 variants, respectively, differed in their migration patterns (Fig. 2a–c). Protein/DNA complexes formed by the FeS cluster-deficient variants migrated further in the gel (Fig. 2a–c). Although it may seem counter-intuitive at first sight given that the C396S variant eluted at a higher estimated molecular weight (Supplementary Fig. 2a), elution patterns in size exclusion chromatography depend on the molecular weight and the shape of the protein, while in native gel EMSAs the exposed charge of the protein plays an additional role. It is hence possible that the exposed charge of DNA2 changes upon FeS cluster loss, which leads to the increased migration observed in EMSAs (Fig. 2a–c). In line with this idea, in Native-PAGE where protein size, shape and charge play a role, wild-type DNA2 and the C396S variant migrate very similarly (Supplementary Fig. 2c), raising the possibility that the presumably more elongated shape of the C396S variant, which should result in a more retarded migration in Native-PAGE, may be counterbalanced by an exposed charge that increases its mobility.

To further investigate the possibility that loss of the FeS cluster induces a conformational change in DNA2, we performed limited proteolysis experiments using Proteinase K. In our assay conditions, full-length wild-type DNA2 and FeS cluster-deficient variants were digested to a similar extent (Fig. 2d and Supplementary Fig. 2d), further suggesting that loss of the FeS cluster does not have a major effect on protein stability. This is different from the related bacterial AddAB where the FeS cluster-deficient variant was quickly degraded, whereas the wild-type enzyme was mostly resistant to trypsin cleavage[15]. The proteolytic patterns of wild-type and FeS cluster-deficient DNA2 however differed. Upon digestion of wild-type DNA2, a stable fragment of ~90 kDa appeared, which was not present in the FeS cluster-deficient variants (Fig. 2d and Supplementary Fig. 2d). Mass spectrometry analysis confirmed that this fragment belonged to DNA2.

**FeS cluster loss affects binding to shorter DNA substrates**. Our results so far suggest that loss of the FeS cluster likely induces a conformational change in DNA2, which affects the sensitivity to proteolytic cleavage, the elution from a size exclusion column, and the migration of DNA2/DNA complexes in EMSAs. To further investigate this possibility, we tested the ability of DNA2 C396S to bind DNA substrates of different ssDNA lengths. We reasoned that a conformational change that affects the overall shape of the protein would likely distort the DNA-binding tunnel, which spans over the nuclease, β-barrel and helicase domains[25].

Zhou and colleagues showed that mouse Dna2 binds with highest affinity to a 5′ overhang substrate with a ssDNA length of 17 deoxythymidine nucleotides (nt)[25]. We therefore used 5′ overhang DNA substrates with single-stranded portions of 10–40 nucleotides. To achieve comparable DNA binding between wild-type DNA2 and DNA2 C396S, we used a 10-fold higher protein concentration for DNA2 C396S. We observed partial DNA binding by wild-type DNA2 for substrates with a minimal overhang of 15 nucleotides and complete binding for substrates with overhangs of 20 and more nucleotides (Fig. 3a, c). In contrast, the C396S variant displayed little binding to substrates with 15 or 20 nucleotides overhangs, and complete binding could only be observed with overhangs of more than 35 nucleotides (Fig. 3b, c).

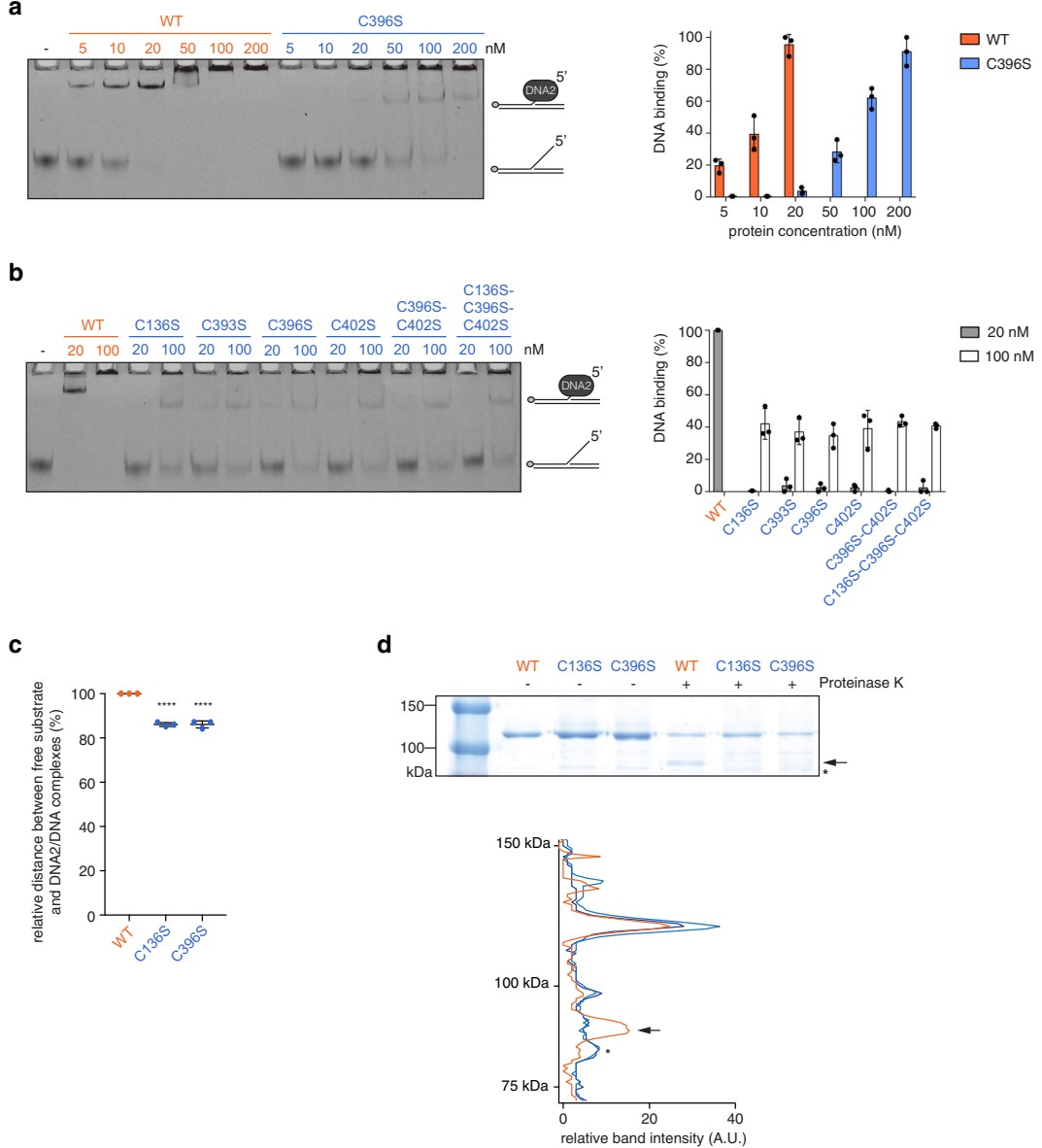

**Fig. 2 FeS cluster loss induces a conformational change that affects DNA binding. a** Representative EMSA of DNA2/DNA complexes using a 5′ flap DNA substrate (FAM-labelled) at 4 nM final concentration and increasing amounts of DNA2 wild-type and C396S, and quantification of DNA binding expressed in % ($n = 3$ independent experiments; error bars, SD). **b** EMSAs as in **a**, using 20 and 100 nM of DNA2 wild-type and FeS cluster-deficient variants ($n = 3$ independent experiments; error bars, SD). **c** Relative distance between free DNA and DNA2/DNA complexes in EMSAs shown in **b**, where distance for wild-type DNA2 samples is set to 100% ($n = 3$ independent experiments; error bars, SD). Statistical analysis: ordinary one-way ANOVA (****$p < 0.0001$). **d** Purified DNA2 wild-type, C136S and C396S were incubated with Proteinase K for 1 min. The reactions were stopped and visualised by SDS–PAGE and InstantBlue staining (top). The band intensities were analysed by a line scan (bottom). The position of the proteolytic product unique to digested wild-type DNA2 is indicated by the arrow. The asterisk represents a heat shock protein contaminant, identified by mass spectrometry. The full gel can be found in Supplementary Fig. 2d.

Taken together, our findings suggest that FeS cluster-deficient DNA2 has an overall reduced affinity for DNA and binds in particular less efficiently to shorter DNA substrates, likely due to a conformational change affecting the DNA-binding tunnel. This is different from what was reported for yeast Dna2 where loss of the FeS cluster did not induce any major structural changes nor affect DNA binding[24]. It is noteworthy that—while the nuclease and helicase domains are highly conserved between yeast and human DNA2—the two proteins differ substantially at the N-terminus with yeast Dna2 containing an additional, 300 amino acids-long, regulatory domain[31,32] that was reported to be able to

bind DNA[33]. This additional DNA-binding site in yeast Dna2 possibly masks any DNA-binding defects associated with FeS cluster loss.

**The FeS cluster is required for all biochemical activities**. We then investigated which biochemical activities of human DNA2 require an intact FeS cluster. To this end, we first performed nuclease assays using both fluorescently labelled (Fig. 4a) and radioactively labelled (Supplementary Fig. 3a) DNA substrates. In contrast to wild-type DNA2, which showed robust nuclease activity at low nanomolar concentrations, FeS cluster-deficient

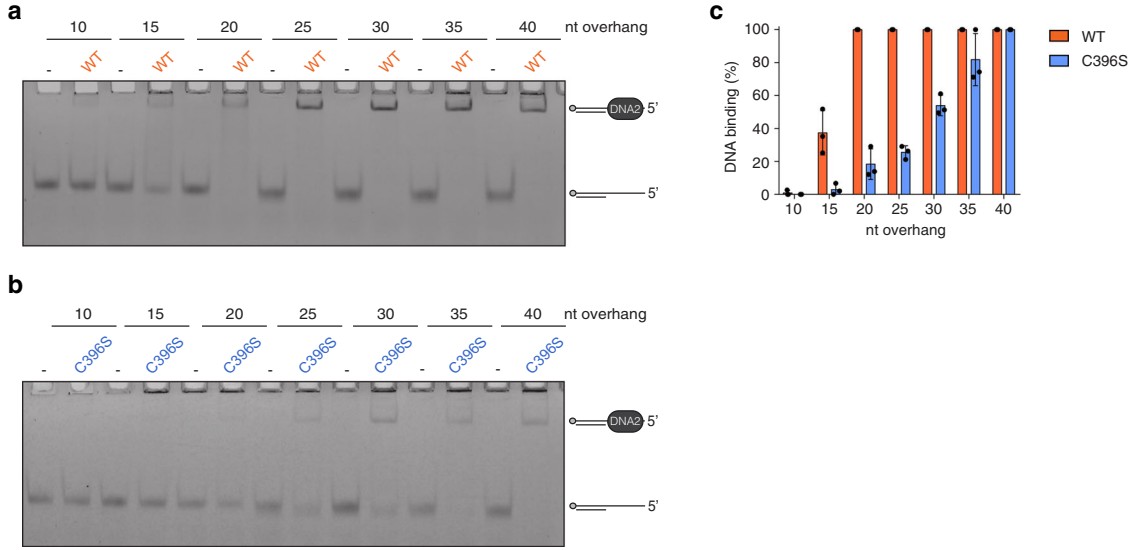

**Fig. 3 FeS cluster loss affects DNA2's ability to grip shorter DNA substrates. a** Representative EMSA of wild-type DNA2/DNA complexes using 5′ overhang substrates containing ssDNA of different lengths. Final protein concentration is 20 nM, final substrate concentration 4 nM. **b** Representative EMSA of DNA2 C396S/DNA complexes using 5′ overhang substrates containing ssDNA of different lengths. Final protein concentration is 200 nM, final substrate concentration 4 nM. **c** Quantification of DNA binding from **a** and **b**, expressed in % ($n = 3$ independent experiments; error bars, SD).

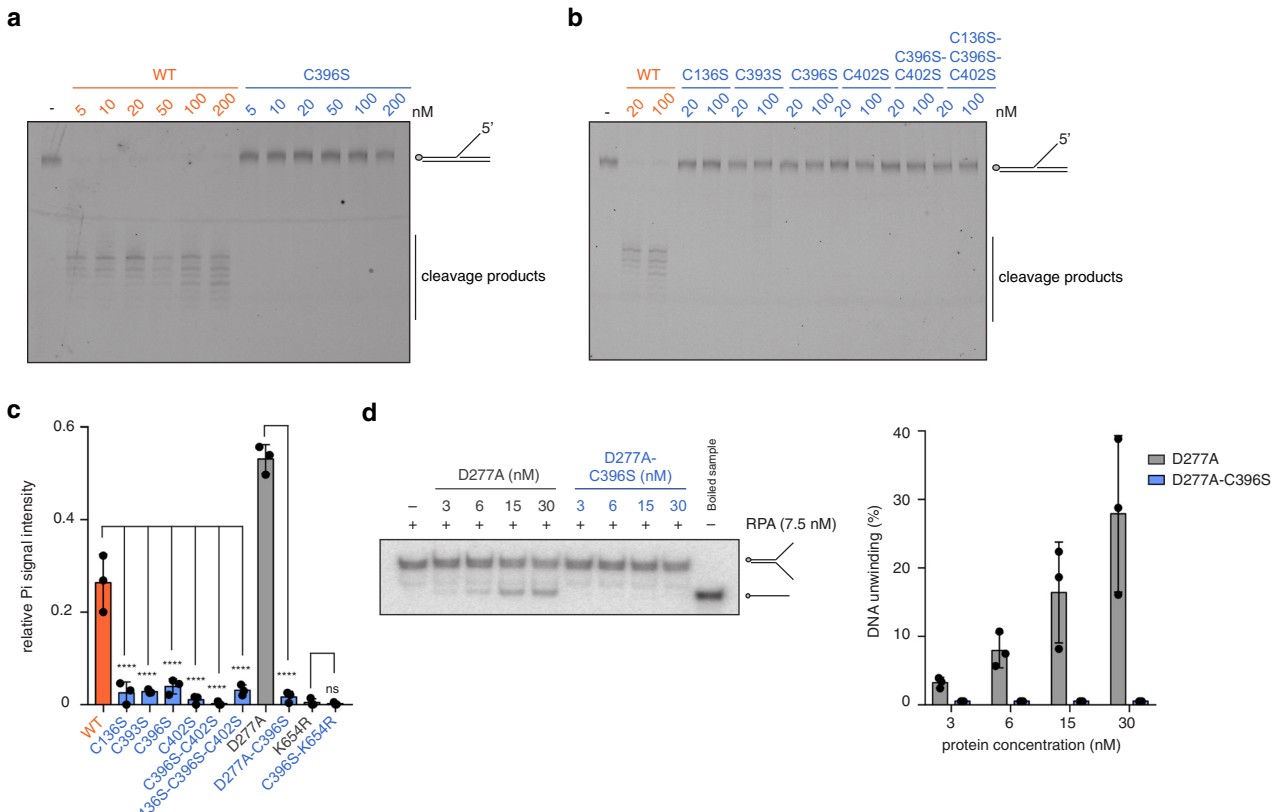

**Fig. 4 The FeS cluster is important for DNA2's nuclease, helicase and ATPase activities. a** Representative nuclease assay showing DNA degradation products generated by DNA2 using a 5′ flap DNA substrate (FAM-labelled) at 4 nM final concentration and increasing amounts of DNA2 wild-type and C396S. **b** Nuclease assays as in **a**, using 20 and 100 nM of DNA2 wild-type and FeS cluster-deficient variants. **c** Quantification of ATPase activity using 0.033 μM γ-$^{32}$P-ATP for DNA2 wild-type, nuclease-deficient D277A, helicase-deficient K654R and FeS cluster-deficient variants at 100 nM final protein concentration. The signal intensity of the released inorganic phosphate (Pi) is measured, and background activity in the absence of DNA is subtracted ($n = 3$ independent experiments; error bars, SD). Statistical analysis: ordinary one-way ANOVA (****$p < 0.0001$; ns, $p = 0.8910$). A representative raw image is shown in Supplementary Fig. 3b. **d** Representative helicase assay showing DNA unwinding using a Y-structure DNA substrate at 1 nM final concentration, 7.5 nM RPA and increasing amounts of DNA2 wild-type and C396S, and quantification of helicase activity expressed in % ($n = 3$ independent experiments; error bars, SD).

DNA2 C396S was unable to cleave the DNA substrate (Fig. 4a and Supplementary Fig. 3a), even at high protein concentrations where we observe DNA binding (Fig. 2a). Likewise, none of the other FeS cluster-deficient variants displayed any nuclease activity (Fig. 4b), similar to what was shown for yeast Dna2[24].

We next wondered whether loss of the FeS cluster also affects the distant helicase domain and checked ATP hydrolysis, which is required for helicase activity and translocation along ssDNA[28,30]. Similar to yeast Dna2[24], we observed reduced DNA-dependent ATPase activity for the FeS cluster-deficient DNA2 variants at protein concentrations where we observe DNA binding (Fig. 4c and Supplementary Fig. 3b). As expected, the nuclease-deficient variant of DNA2 (D277A), which cannot degrade the DNA substrate that functions as a co-factor, displayed a higher apparent ATPase activity than the wild-type enzyme (Fig. 4c and Supplementary Fig. 3b), an effect already reported elsewhere[29].

To study directly the effect of FeS cluster loss on helicase activity, we then used DNA2 D277A (to prevent degradation of the substrate) in a helicase assay employed previously[29]. First, we verified that DNA2 D277A and its FeS cluster-deficient counterpart DNA2 D277A-C396S had similar DNA-binding affinities as DNA2 WT and C396S, respectively (Supplementary Fig. 3c, d), and that we could observe residual DNA binding for DNA2 D277A-C396S at the protein concentration range used previously[29]. As for ATP hydrolysis, FeS cluster loss abrogated the helicase activity of DNA2 even at concentrations where we observe DNA binding, since we could observe up to 30% DNA unwinding for DNA2 D277A, but no helicase activity for the FeS cluster-deficient variant (Fig. 4d).

In summary, the FeS cluster in DNA2 is required for all of its biochemical activities, including nuclease, helicase and ATPase activities. Importantly, FeS cluster-deficient variants did not display any activity even at high protein concentrations where they are able to bind DNA, suggesting that the observed defects are not only a consequence of impaired DNA binding.

**Oxidation affects DNA binding but not FeS cluster binding**. While our data so far suggest an important structural role for the FeS cluster in DNA2, we nevertheless wondered whether it could also have a redox role, similar to what was reported for the bacterial helicase DinG[14]. Interestingly, incubation of wild-type DNA2 with increasing amounts of hydrogen peroxide ($H_2O_2$), an oxidising agent, gradually impaired DNA binding (Fig. 5a). This effect was reversible upon addition of the reducing agents dithiothreitol (DTT) (Fig. 5a) and tris(2-carboxyethyl)phosphine (TCEP) (Supplementary Fig. 4a), indicating that DNA2 may be redox-sensitive. We also observed this effect using yeast Dna2 (Supplementary Fig. 4b), suggesting an evolutionarily conserved function. It is of note that the $H_2O_2$ concentrations used in our assays do not appear to generally affect DNA-binding proteins, since Helicase-like transcription factor (HLTF) was not sensitive to $H_2O_2$ treatment in DNA-binding assays (Supplementary Fig. 4c). Importantly, when DNA2 was bound to DNA first, its oxidation sensitivity was greatly reduced (Fig. 5b), suggesting that —once DNA2 is in a DNA-bound conformation—any redox-sensitive residue(s) are locked in place or shielded.

We next investigated whether this oxidation sensitivity was FeS cluster-dependent. Replacing any of the four FeS cluster-ligating cysteines, however, did not abolish the observed redox sensitivity (Fig. 5c). We also included the double and triple cysteine-to-serine variants to prevent the formation of disulphide bridges among the cysteine residues that normally ligate the FeS cluster. The presence of additional disulfide bridges was for example observed in Fumarate nitrate reductase regulator (FNR), a

bacterial $O_2$ sensor, upon oxidation and loss of its FeS cluster[34,35]. The double and triple variants were however as sensitive to oxidation by $H_2O_2$ and subsequent reduction by DTT as the single variants (Fig. 5c), suggesting that the FeS cluster in DNA2 is not responsible for the redox sensitivity observed in DNA-binding assays.

To avoid using an oxidising agent, we also incubated DNA2 under non-reducing conditions for atmospheric oxygen to oxidise it. After 20 h we tested its DNA-binding activity, which was severely impaired, but fully recovered upon DTT treatment (Fig. 5d). To verify that under these conditions the FeS cluster remains intact, we set out to perform UV–visible (UV–vis) spectroscopy on purified mouse Dna2, since we could not obtain a sufficiently concentrated sample of human DNA2. First, we confirmed that purified mDna2 (Supplementary Fig. 4d) binds to DNA with a similar affinity than human DNA2 (Supplementary Fig. 4e) and loses its ability to bind DNA when incubated for 20 h at atmospheric oxygen (Fig. 5e). As for human DNA2, DNA binding by mDna2 could be fully recovered upon treatment with a reducing agent (Fig. 5e).

In UV–vis spectroscopy, mDna2 displayed a shoulder at 410 nm, indicative of the presence of an FeS cluster (Fig. 5f and Supplementary Fig. 4f). Importantly, the spectra did not change when mDna2 was left to oxidise for 20 h or when re-reduced by TCEP after oxidation (Fig. 5f and Supplementary Fig. 4f), confirming that the observed oxidation sensitivity of DNA2/mDna2 is not mediated by its FeS cluster.

Taken together, our data establish that DNA binding by DNA2 is sensitive to oxidation by hydrogen peroxide treatment and air exposure. Unexpectedly, however, we find no signs of FeS cluster oxidation, suggesting that other, so far unknown, residues in DNA2 must act as redox sensors.

## Discussion
In this work, we show that the FeS cluster in human DNA2 is an important structural element for all of DNA2's biochemical activities. Substituting any of the FeS cluster-coordinating cysteines leads to FeS cluster loss, and introducing multiple substitutions does not affect DNA2's biochemical activities further, indicating that all four cysteines are essential for the structural integrity of the FeS cluster-containing pocket. However, FeS cluster loss does not completely abrogate protein stability. Moreover, in our biochemical assays, FeS cluster-deficient DNA2 —while displaying reduced DNA-binding affinity—was still able to bind DNA at high protein concentrations. Thus, FeS cluster-deficient variants may still be able to retain some of DNA2's cellular functions. On the other hand, the nuclease, ATPase and helicase activities were severely affected upon FeS cluster loss, even in conditions where the protein could bind DNA. The cysteine-to-serine DNA2 variants, which still interact with the CIA targeting complex in cells and therefore represent the apo-protein, provide the tools to study the effect of FeS cluster loss in a cellular context.

Interestingly, we observed that the protein/DNA complexes formed by wild-type and FeS cluster-deficient variants exhibited different migration patterns in EMSAs, an effect not seen in other FeS cluster-containing DNA-binding proteins[15,36]. We could further show by size exclusion chromatography, limited proteolysis, and DNA-binding assays with shorter DNA substrates that this migration effect is likely caused by a conformational change upon FeS cluster loss. Somewhat surprisingly, such an altered migration pattern was not observed with yeast Dna2[24], although we cannot exclude that the effect could have been masked in EMSAs due to the larger size of the yeast protein and the presence of an additional DNA-binding module[33]. Taking into account

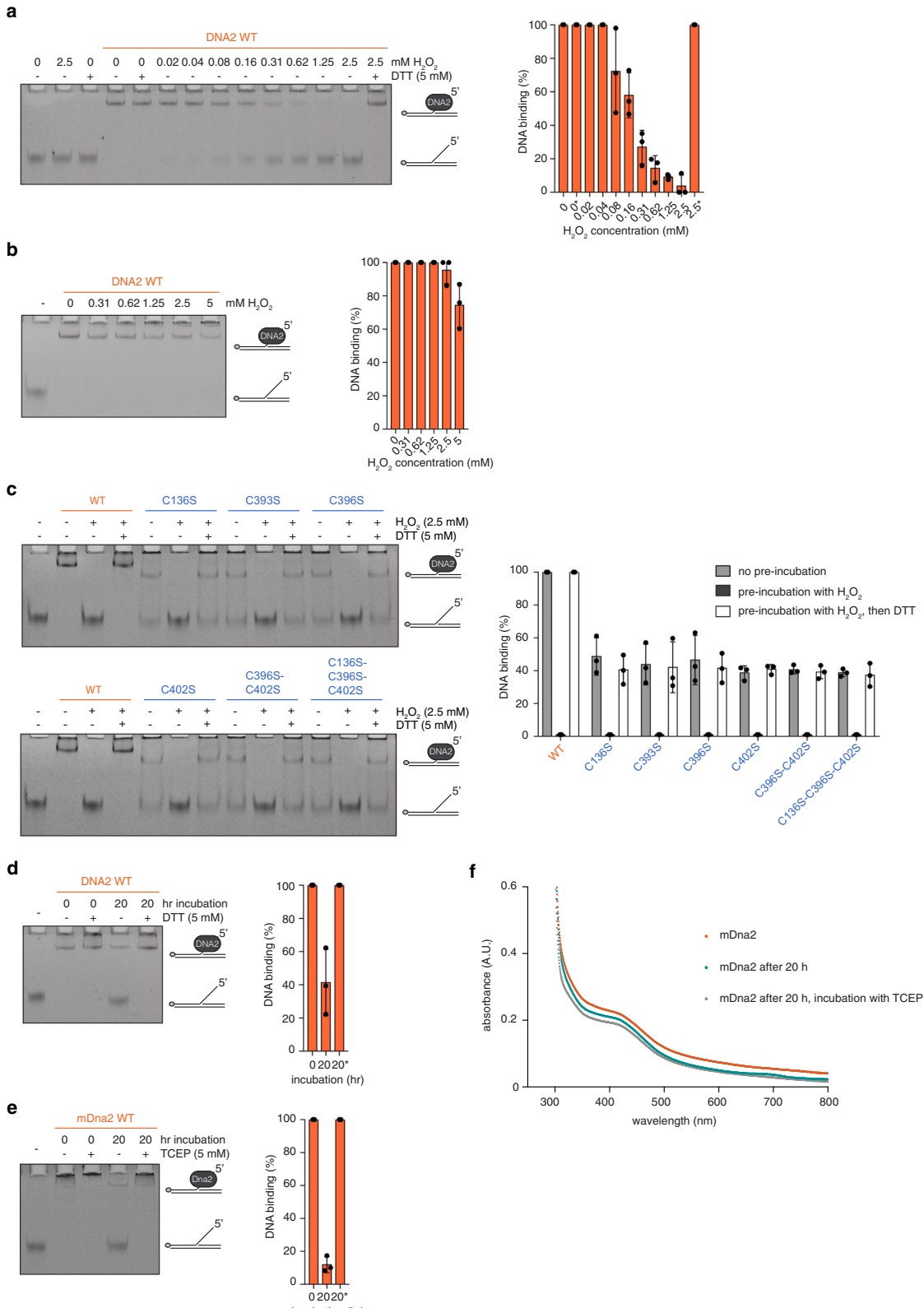

that FeS cluster loss did neither induce any major structural changes nor affect DNA binding in yeast Dna2[24], it would however appear that FeS cluster loss affects human DNA2 more severely than yeast Dna2. In light of our results, high-resolution structural studies of the FeS cluster-deficient DNA2/Dna2 will be required to understand the differences between the human and yeast proteins. Such studies should also help to elucidate how

mechanistically the nuclease and helicase activities of DNA2 are coupled and define the function of the FeS cluster in this process.

While investigating a potential redox role for the FeS cluster, we discovered that DNA2 is sensitive to oxidation by air and hydrogen peroxide, and that this effect is reversible upon reduction. Surprisingly, all FeS cluster-deficient cysteine variants were also redox-modulated, implying that the FeS cluster in

**Fig. 5 Oxidised DNA2 loses its ability to bind DNA, while maintaining a stably bound FeS cluster. a** Representative EMSA of DNA2/DNA complexes upon pre-incubation with increasing amounts of $H_2O_2$. DTT was added after the $H_2O_2$ treatment. The DNA-binding reaction was set up after $H_2O_2$ and DTT treatment. Final protein concentration is 50 nM and final 5′ flap DNA substrate concentration is 4 nM. Quantification of DNA binding expressed in %. Asterisk indicates addition of DTT ($n = 3$ independent experiments; error bars, SD). **b** Representative EMSA of DNA2/DNA complexes where the DNA-binding reaction was incubated for 30 min before the addition of increasing amounts of $H_2O_2$. Final protein concentration is 50 nM and final 5′ flap DNA substrate concentration is 4 nM. Quantification as in **a** ($n = 3$ independent experiments; error bars, SD). **c** Representative EMSA of DNA2/DNA complexes upon pre-incubation with $H_2O_2$ using 50 nM of wild-type DNA2 and 100 nM of FeS cluster-deficient variants. DTT was added after the oxidative treatment. The DNA-binding reaction was set up after $H_2O_2$ and DTT treatment. Quantification as in **a** ($n = 3$ independent experiments; error bars, SD). **d** Representative EMSA of DNA2/DNA complexes after 20 h incubation without reducing agents. DTT was added after the incubation and prior to the DNA-binding assay. Quantification as in **a** ($n = 3$ independent experiments; error bars, SD). **e** Representative EMSA of mDna2/DNA complexes after 20 h incubation without reducing agent. TCEP was added after the incubation and prior to the DNA-binding assay. Quantification as in **a** ($n = 3$ independent experiments; error bars, SD). **f** UV–vis spectroscopy with mDna2 at 25 µM, as-purified (orange), after 20 h of air exposure (teal), and after 20 h of air exposure followed by TCEP treatment (grey). The complete spectra are found in Supplementary Fig. 4f.

DNA2 is not responsible for the redox effect observed in DNA-binding assays. Moreover, DNA binding was completely recovered under reducing conditions, indicating that the FeS cluster in DNA2 is stably coordinated, at least in vitro, and does not get irreversibly damaged and lost under oxidising conditions. This notion was further confirmed by UV–vis spectroscopy where mDna2 did not display any signs of FeS cluster oxidation upon 20 h of air exposure. However, the redox chemistry of the FeS cluster might still affect DNA2's functions in more subtle ways that could not be captured in our biochemical assays.

A few decades ago, the tumour suppressor p53[37], and more recently, the transcription factor Brf2[38] and the homologous recombination protein RAD51[39], were found to contain oxygen-reactive cysteines. In the case of the transcription factor Brf2, its redox-reactive cysteine was found to interact with the TATA box, and DNA binding was reversibly impaired upon oxidation of the protein[38]. Redox regulation is therefore an emerging theme in DNA processing. Oxidative modification of cysteine residue(s), either as single entities or as pairs to form disulphide bridges, may modulate DNA2's functions, in addition to ubiquitination and sumoylation that were recently found to regulate human and yeast DNA2/Dna2, respectively[31,40].

Interestingly, we found that when DNA2 is in a DNA-bound conformation, its oxidation sensitivity is greatly reduced, suggesting that any redox-sensitive residue(s) are locked in place or no longer accessible. In light of these results, we wondered whether there was a cysteine residue located in the DNA-binding tunnel, which could act as a redox sensor. Cysteine-743 seemed to be the most likely candidate, since, although it is not directly involved in DNA binding, it is located close to the DNA-binding tunnel[25] and conserved from yeast to human. However, purified DNA2 C743S (Supplementary Fig. 5a, b) displayed a similar oxidation sensitivity as wild-type DNA2 (Supplementary Fig. 5c, d), thus excluding cysteine-743 as a redox-sensitive residue.

Apart from cysteine-743 and the FeS cluster-coordinating cysteines, 25 other cysteine residues in human DNA2 remain as possible redox-sensors. Since yeast and mouse Dna2 also show redox sensitivity in our assays, strictly conserved cysteine residues would seem the most likely candidates for a redox role. However, cysteines which are not conserved in the primary sequence may still perform the same function in three-dimensional space. Moreover, given that a localised change in one of the domains of DNA2 can affect the central DNA-binding tunnel and consequently DNA2's activities, the redox cysteine(s) may be located far away from the tunnel and still regulate DNA binding. The focus of future work will be to identify which of the cysteine(s) are oxidation-sensitive and to investigate the importance of DNA2's redox-sensitivity in living cells.

In summary, our study shows that the FeS cluster in DNA2 primarily functions to support the structure of the polypeptide. It thus contributes to shaping the central DNA-binding tunnel and is required for DNA2's nuclease, helicase and ATPase activities. Although its FeS cluster may not act as a redox sensor, it seems that DNA2 contains another, unidentified redox module regulating DNA binding. Interestingly, it was also shown that cysteines coordinating $Zn^{2+}$ in the 70-kDa subunit of human RPA, which recruits DNA2 to ssDNA[26,27], form disulfide bonds upon oxidation, resulting in reversible loss of the divalent ion and decreased DNA-binding affinity[41–44]. DNA binding by DNA2 may, hence, theoretically be prevented by oxidation of both RPA and DNA2. Whether or not this double-safety mechanism is relevant in vivo and contributes to DNA2's regulation at the molecular level needs to be further investigated.

## Methods

**Plasmids and baculoviruses**. pFastBac1-His₆-DNA2-FLAG[29] allowed for the expression of codon-optimised human DNA2 in Sf9 insect cells. Site-directed mutagenesis was performed based on the QuikChange Site-Directed Mutagenesis (Stratagene) approach. Bacmids and baculoviruses were generated using the Bac-to-Bac Baculovirus Expression System (Invitrogen).

For expression in mammalian cells, human DNA2 cDNA was amplified from pBabe-hygro-3xFLAG DNA2 wt[45]. The cDNA was cloned into pDONR221 GATEWAY entry vector (Invitrogen) according to the manufacturer's protocol. Site-directed mutagenesis was performed based on the QuikChange Site-Directed Mutagenesis (Stratagene) approach. The pDONR221-DNA2 constructs were then cloned into GATEWAY destination vectors according to the manufacturer's protocol.

HLTF cDNA was purchased from the Mammalian Gene Collection (Dharmacon) as bacterial stabs (clone ID: 6015181), cloned into pDONR221 GATEWAY entry vector and shuttled to GATEWAY destination vectors according to the manufacturer's protocol (Invitrogen). Bacmids and baculoviruses were generated using the Bac-to-Bac Baculovirus Expression System (Invitrogen) approach.

**Recombinant protein expression and purification**. Human DNA2 variants were purified from Sf9 insect cells (Invitrogen). Liquid cultures, grown in HyClone SFX-insect medium (GE Healthcare) at a density of $1.5 \times 10^6$ cells/ml, were infected with baculoviruses at a multiplicity of infection (MOI) of 1. Cells were incubated at 25 °C for 48 h shaking. Cells were collected by centrifugation for 30 min at 17,200×g and 4 °C and pellets were lysed in a lysis buffer containing 100 mM NaCl, 50 mM Tris–HCl (pH 8.0), 10% glycerol, 1 mM EDTA, 5 mM ATP, 2.5 mM MgCl₂, 0.1% NP-40, 2 mM DTT, cOmplete EDTA-free Protease Inhibitor Cocktail tablets (Roche) and 0.1% Benzonase (Santa Cruz) (pellet from 1 l of culture resuspended in 20 ml of lysis buffer) for 20 min on ice. Lysates were spun down to remove insoluble material and filtered through 0.45 µm filters. Filtered lysates were incubated with equilibrated anti-FLAG M2 affinity gel beads (Sigma Aldrich) for 1 h at 4 °C with rotation (0.4 ml anti-FLAG M2 beads packed volume incubated with 20 ml lysate). Beads were washed in wash buffer A (100 mM NaCl, 50 mM Tris–HCl (pH 8.0), 10% glycerol, 1 mM EDTA, 5 mM ATP, 2.5 mM MgCl₂, 2 mM DTT; 20 ml of buffer for 0.4 ml anti-FLAG M2 beads packed volume) for 1 h at 4 °C rotating. The beads were then washed twice in wash buffer A (20 ml of buffer for 0.4 ml anti-FLAG M2 beads packed volume) for 15 min at 4 °C with rotation. The three washing steps were repeated with wash buffer B (as wash buffer A but without 5 mM ATP, 2.5 mM MgCl₂). The proteins were then eluted with elution buffer containing 100 mM NaCl, 50 mM Tris–HCl (pH 8.0), 10% glycerol, 1 mM EDTA, 2 mM DTT and 200 ng/µl 3x FLAG peptide (Sigma Aldrich) for 1 h at 4 °C rotating (0.4 ml of elution buffer used for 0.4 ml of packed anti-FLAG M2 beads

volume). FLAG-IP eluates were aliquoted, flash-frozen in liquid nitrogen and stored at −80 °C. Protein concentration was estimated by SDS–PAGE analysis and InstantBlue staining, based on a BSA standard curve.

Mouse Dna2 was expressed in *Hi5* insect cells as an N-terminally hexa-histidine-tagged full-length protein using baculoviruses derived from pAc-His$_6$-TEV-*mDna2* (gift from Susanne Kassube). Cells were grown for 40 h, as described for human DNA2, and lysed in a lysis buffer containing 200 mM NaCl, 50 mM Tris–HCl (pH 8.0), 5% glycerol, 0.1% NP-40, 1 mM TCEP, cOmplete EDTA-free Protease Inhibitor Cocktail tablets (Roche) (pellet from 1 l of culture resuspended in 50 ml of lysis buffer) by sonication in a Sonopuls GM70 (Bandelin). Lysates were spun down to remove insoluble material and filtered through 0.45 μm filters. Filtered lysates were incubated with equilibrated TALON Metal Affinity resin (Clontech) for 1 h at 4 °C with rotation (8 ml TALON resin-packed volume incubated with 150 ml lysate). Beads were washed in wash buffer A (200 mM NaCl, 50 mM Tris–HCl (pH 8.0), 5% glycerol, 2 mM beta-mercaptoethanol; 50 beads packed volume), wash buffer B (500 mM NaCl, 50 mM Tris–HCl (pH 8.0), 5% glycerol, 2 mM beta-mercaptoethanol; 20 beads packed volume) and wash buffer C (100 mM NaCl, 50 mM Tris–HCl (pH 8.0), 5% glycerol, 2 mM beta-mercaptoethanol, 4 mM imidazole (pH 7.0); 20 beads packed volume). The proteins were then eluted with elution buffer containing 100 mM NaCl, 50 mM Tris–HCl (pH 8.0), 5% glycerol, 2 mM beta-mercaptoethanol and 50 mM imidazole (pH 7.0). The eluted sample was diluted 1:1 with no-salt buffer (50 mM Tris–HCl (pH 8.0), 5% glycerol, 2 mM DTT) and loaded onto a 5 ml HiTrap Heparin HP (GE healthcare). The proteins were eluted with a linear NaCl gradient (from 80 to 1 M NaCl) in a buffer also containing 50 mM Tris–HCl (pH 8.0), 5% glycerol, 2 mM DTT. Fractions corresponding to mDna2 were pooled, diluted to 100 mM NaCl with no-salt buffer and concentrated using Vivaspin 6, 100K (Sartorius). The purified protein was aliquoted and flash-frozen in liquid nitrogen.

Yeast Dna2 was expressed from a modified pGAL:*DNA2* vector[46] containing an N-terminal FLAG/HA-tag and a C-terminal His$_6$ tag in the *S. cerevisiae* strain WDH668, and purified as described previously[47]. Briefly, lysates were subjected to Ni$^{2+}$-NTA and anti-FLAG M2 pull-downs and proteins were eluted in a buffer containing 25 mM Tris–HCl (pH 7.5), 150 mM NaCl, 10% glycerol, 1 mM β-mercaptoethanol and 200 ng/μl 3x FLAG peptide (Sigma Aldrich).

Human HLTF was purified from *Sf9* insect cells. Cells were grown as described above and lysed in five packed-cell volumes of lysis buffer (50 mM Na-phosphate pH 7.0, 150 mM NaCl, 10% glycerol, 0.1% NP-40, 1 mM TCEP, 0.5 mM EDTA), supplemented with protease inhibitors cocktail (Roche) for 30 min on ice. The lysate was centrifuged at 4 °C in a Sorval WX+ (Thermo Scientific) ultracentrifuge equipped with a T-865 rotor at 100,000×g for 1 h. The supernatant was filtered through 0.22 μm filters and incubated with 0.01 volumes of equilibrated anti-FLAG M2 beads (Sigma Aldrich) for 2 h at 4 °C rotating. The beads were then washed three times for 10 min in 10 ml of lysis buffer at 4 °C with rotation, and three times with the storage buffer (50 mM Na-phosphate pH 7.0, 300 mM NaCl, 10% glycerol, 1 mM TCEP). Bound proteins were eluted with five beads volumes of the storage buffer supplemented with 200 ng/μl 3x FLAG peptide (Sigma Aldrich) for 1 h at 4 °C rotating. Eluates were pooled, filtered on 0.22 μm spin columns (Bio-Rad), concentrated on Amicon Ultra filters (Merck Millipore) and loaded on a Superdex 200 10/300 GL size-exclusion chromatography column (GE Healthcare). Fractions containing HLTF were aliquoted, flash-frozen in liquid nitrogen and stored at −80 °C.

Human RPA was purified from *E. coli* through HiTrap Blue, HiTrap Desalting and HiTrap Q columns (GE Healthcare) according to a previously described protocol[48].

**Iron incorporation assay.** 20 ml liquid cultures of *Sf9* cells at $1.5 \times 10^6$ cells/ml, supplemented with 200 μl of 0.1 M sodium ascorbate and 20 μl of $^{55}$FeCl$_3$ (1 mCi/ml, PerkinElmer), were infected with baculoviruses at an MOI of 1 and incubated at 25 °C for 48 h. Cells were then centrifuged at 500×g for 10 min at 4 °C. The cell pellets were first washed with 5 ml of citrate buffer (50 mM citrate and 1 mM EDTA in PBS, pH 7.0) and then with 10 ml PBS. The pellets were resuspended in 1 ml of lysis buffer (as above, for DNA2). The lysates were incubated on ice for 30 min and then centrifuged for 30 min at 17,200×g and 4 °C. Next, the lysates were incubated with 20 μl of pre-equilibrated anti-FLAG M2 affinity gel (Sigma Aldrich) for 1 h at 4 °C rotating. The beads were then washed three times with 1 ml of wash buffer A (as above) and three times with wash buffer B (as above). The beads were then incubated with 100 μl of elution buffer (as above) for 1 h at 4 °C. 1 ml of Ultima Gold liquid scintillation liquid (PerkinElmer) and 90 μl of the eluted proteins were added to scintillation counting tubes, which were then vortexed for mixing. Counts per minute (cpm) were measured with a Tri-Carb scintillation counter (Packard) using standard $^3$H settings. The remaining 10 μl elutions were analysed by SDS–PAGE and InstantBlue staining. The counts were normalised based on protein levels.

**Native-PAGE.** 60, 120 and 180 ng of purified DNA2 WT and C396S were mixed with 4x sample buffer (62.5 mM Tris–HCl, pH 8.8, 25% glycerol, bromophenol blue), loaded onto a 6% native polyacrylamide gel, and run in cold running buffer (192 mM glycine, 25 mM Tris–HCl pH 8.3) at 100 V for 1h30. The samples were analysed by SDS–PAGE and Western blotting.

**Limited proteolysis.** 225 ng of purified DNA2 WT, C136S and C396S were incubated with 2 ng of Proteinase K (Thermo Fisher Scientific) for 1 min at 37 °C. The 10 μl reactions were stopped by addition of 2.5 μl 5x SDS sample loading buffer (250 mM Tris–HCl pH 6.8, 10% SDS, 30% glycerol, 10 mM DTT, bromophenol blue) and boiling at 95 °C for 5 min. The samples were analysed by SDS–PAGE and InstantBlue staining.

**Gel filtration.** Purified DNA2 WT and C396S were subjected to size-exclusion chromatography on a Superdex 200 Increase 10/300 GL (GE Healthcare) equilibrated in 100 mM NaCl, 50 mM Tris–HCl (pH 8.0), 10% glycerol, 1 mM EDTA and 2 mM DTT.

**DNA substrates.** Oligonucleotides used in this study (Supplementary Table 1) were synthesised by Microsynth, either unlabelled or labelled with fluorescein amidite (FAM) at the 3′- or 5′-end. Fluorescently labelled DNA substrates (Supplementary Table 2) were generated by annealing 200 nM of FAM-labelled oligonucleotide and 200–400 nM of unlabelled oligonucleotides in 10 mM Tris–HCl (pH 8.0), 50 mM NaCl, and 10 mM MgCl$_2$ in a PCR cycler, where samples were heated to 95 °C for 5 min, then cooled down by 5 °C every 3 min to 20 °C. For the EMSA using the radioactively labelled Y-structure DNA substrate (Supplementary Table 3), 250 nM of X04 was $^{32}$P-labelled at the 3′-end with [alpha-$^{32}$P] cordy-cepin-5′ triphosphate (Perkin Elmer) and terminal transferase (New England Biolabs) according to the manufacturer's instructions. Unincorporated nucleotides were removed with Illustra MicroSpin G-25 columns (GE Healthcare). 50 nM of 3′-labelled X04 were annealed in 10 mM Tris–HCl (pH 8.0), 50 mM NaCl, and 10 mM MgCl$_2$ by heating to 95 °C for 5 min, letting the samples cool down to room temperature and putting them to 4 °C. For the nuclease assay using the $^{32}$P-labelled 5′ flap DNA substrate (Supplementary Table 3), 250 nM of X04 was labelled as above, and 50 nM of 3′-labelled X04 was annealed with 75 nM of unlabelled X01 and 100 nM of unlabelled X02.1/2. For the helicase assay using the radioactively labelled large Y-structure (Supplementary Table 3), 300 nM of X12-3HJ3 was $^{32}$P-labelled at the 3′-end with [alpha-$^{32}$P] dCTP (Perkin Elmer) and terminal transferase (New England Biolabs) according to the manufacturer's instructions. Unincorporated nucleotides were removed with Micro Bio-Spin P-30 Tris chromatography columns (Biorad). 150 nM of 3′-labelled X12-3HJ3 was annealed with a two-fold excess of unlabelled X12-3TOPL in terminal transferase buffer (New England Biolabs) by heating to 95 °C for 3 min and letting the samples cool down to room temperature overnight.

**EMSA.** The reactions were set up in 10 μl in a buffer containing 25 mM Tris–HCl (pH 8.0), 0.1 mg/ml BSA, 5 mM EDTA and with or without 1 mM DTT. For the EMSAs using fluorescently labelled DNA substrates, the complexes were assembled by mixing 50–2500 fmol of purified proteins (corresponding to 5–200 nM final protein concentrations, as indicated in the figures) with 40 or 50 fmol of FAM-labelled DNA substrates (final concentration of 4 or 5 nM, respectively) and incubated on ice for 30 min. For the EMSAs with H$_2$O$_2$ and DTT, 0.5 or 1 pmol of purified protein diluted in purification buffer without DTT were pre-incubated with 0.2–25 nmol H$_2$O$_2$ for 30 min on ice, then with or without 50 nmol DTT for 30 min on ice. The DNA-binding reaction was performed as above. For the EMSAs using the radioactively labelled Y-structure DNA substrate, 1 fmol of radioactively labelled 5′-flap substrate (corresponding to 1 nM final concentration) and 6–60 fmol of protein (corresponding to 3–30 nM final protein concentrations) were used. 10 μl of 2x sample loading dye (30% glycerol, 40 mM Tris–HCl pH 7.6, 0.5 mM EDTA, bromophenol blue) were added to the reactions and the protein–DNA complexes were separated on a 10% non-denaturing polyacrylamide gel (1x Tris–acetate–EDTA (TAE), 10% acrylamide 19:1) in cold TAE buffer at 80 V for 2 h. For the EMSAs with yeast Dna2 and HLTF, the protein/DNA complexes were separated on a 5% non-denaturing polyacrylamide gel (0.5x Tris–Borate–EDTA (TBE), 5% acrylamide 19:1) in 0.5x TBE buffer at 80 V for 1 h. For fluorescence signal detection, the gels were scanned on a Typhoon FLA9500 laser scanner (GE Healthcare) with the fluorescence-imaging setting. For radioactive signal detection, the gels were exposed to storage phosphor screens (GE Healthcare) prior to scanning using the phosphor-imaging setting. Uncropped gels can be found in Supplementary Fig. 6.

**Nuclease assay.** 10 μl reactions containing 50 mM Tris–HCl (pH 8.0), 5 mM MgCl$_2$, 0.1 mg/ml BSA, 1 mM DTT, 5–200 nM of purified proteins (as indicated in the figures) and either 1 nM of $^{32}$P-labelled or 4 nM of FAM-labelled 5′ flap DNA substrate were incubated at 37 °C for 30 min. Reactions were stopped by adding 10 μl of stop solution (10 mM EDTA, 96% formamide, bromophenol blue) and boiling at 95 °C for 10 min. Samples were loaded on a 20% denaturing poly-acrylamide gel (1x TBE, 7 M urea, 20% acrylamide 19:1) and run in 1x TBE buffer at 10–18 W for 2 h. For fluorescence signal detection, the gels were scanned on a Typhoon FLA9500 laser scanner (GE Healthcare) with the fluorescence-imaging setting. For radioactive signal detection, the gels were exposed to storage phosphor screens (GE Healthcare) prior to scanning using the phosphor-imaging setting. Uncropped gels can be found in Supplementary Fig. 6.

**ATPase assay**. 10 µl reactions containing 5 mM MgCl$_2$, 0.01 mM ATP, 0.033 µM γ-$^{32}$P-ATP, 5–200 nM of purified proteins (as indicated in the figures), either with or without 50 nM of 5′ flap DNA substrate, were incubated at 37 °C for 30 min. The reactions were stopped by addition of EDTA (50 mM final concentration). 1 µl of the samples were spotted on a PEI-Cellulose TLC plate (Merck), and the plates were resolved in a solution containing 0.15 M LiCl and 0.15 M formic acid. Plates were then air-dried, wrapped in cling film and exposed to a storage phosphor screen (GE Healthcare). The signal was detected on a Typhoon FLA9500 scanner (GE Healthcare).

**Helicase assay**. Helicase assays were performed as previously described[29]. Briefly, 15 µl reactions were set up using 1 nM of DNA substrate (3′-labelled large Y-structure) and the indicated amounts of purified proteins in a buffer containing 25 mM Tris–acetate pH 7.5, 5 mM magnesium acetate, 1 mM ATP, 1 mM DTT, 0.1 mg/ml BSA, 1 mM phosphoenolpyruvate, 80 units/ml pyruvate kinase and 50 mM NaCl. Reactions were incubated at 37 °C for 30 min and stopped by adding 5 µl 2% stop solution (150 mM EDTA, 2% SDS, 30% glycerol, bromophenol blue) supplemented with 1 µl Proteinase K (20 mg/ml, Roche) and a 20-fold excess of the cold oligonucleotide (with the same sequence as the radioactively labelled one) and incubated at 37 °C for 10 min. The samples were analysed on a 10% non-denaturing polyacrylamide gel (1x TBE, 10% acrylamide 19:1) in TBE buffer. The gels were dried on 17 CHR chromatography paper (Whatman) and exposed to a storage phosphor screen (GE Healthcare). The signal was detected on a Typhoon FLA9500 scanner (GE Healthcare). Uncropped gels can be found in Supplementary Fig. 6.

**Quantification of biochemical assays**. Image quantification was done in Image J (version 2.0.0-rc-43/1.52c)[49]. For EMSAs, the percentage of substrate shifted relative to the total substrate (unshifted and shifted DNA) was quantified. If the DNA/protein complex was fully shifted to the wells, the data was not quantified (i.e. not set to 100%) since this effect could reflect non-specific assemblies due to high protein concentration. For nuclease assays, the percentage of DNA substrate disappearing, relative to the substrate alone (without protein), was measured. For ATPase assays, the signal intensity of the released inorganic phosphate (Pi) relative to the total signal (remaining ATP and released Pi) was quantified for all samples in the absence or presence of DNA. The background activity in the absence of DNA was then subtracted. Finally, for helicase assays, the percentage of DNA substrate unwound relative to the total substrate (double-stranded and unwound DNA) was measured.

**Statistics and reproducibility**. Unless stated otherwise, biochemical assays were repeated three times with largely consistent results. In the graphs, the mean ± standard deviation (SD) is displayed. The exact number of independent experiments, from which the mean is calculated, is given in the legend of each figure. The statistical analyses were based on an ordinary one-way ANOVA test. The exact p-values are indicated in the figure or figure legend, unless $p < 0.0001$.

**UV–vis spectroscopy**. UV spectra were collected on a Cary 60 UV–vis spectrometer (Agilent) with Cary WinUV 5.1 software. Measurements were performed over a spectral range of 200–800 nm at a scan speed of 300 nm/min with 0.5 nm data intervals. The buffer solution baseline was automatically subtracted.

After collection of the first spectrum, mDna2 at 25 µM was incubated at 4 °C for 20 h, in a similar setup to the one used in EMSAs. After the collection of the spectrum after 20 h, TCEP was added to a final concentration of 57 mM and incubated for 30 min.

**Mammalian cell culture**. HEK293T (ATCC CRL-3216) cells were cultured in DMEM containing 4.5 g/L D-glucose, L-glutamine, pyruvate (Gibco) and supplemented with 10% foetal calf serum (FCS) in a humidified incubator at 37 °C and 6% CO$_2$ atmosphere.

Parental HeLa Flp-In T-REx cells (gift from Stephen S. Taylor) were cultured in DMEM as above, supplemented with 10% FCS, 15 µg/ml blasticidin and 100 µg/ml zeocin (LabForceAG). Flp-In-compatible plasmids containing the gene of interest were co-transfected with a plasmid coding for Flp recombinase (pOG44) using Lipofectamine 2000 (Thermo Fisher Scientific) according to the manufacturer's instructions. After integration of the expression constructs, the DMEM medium was supplemented with 10% FCS, 15 µg/ml blasticidin and 150 µg/ml hygromycin B (LabForce AG).

**DNA2–CIA targeting complex co-immunoprecipitation**. HEK293T cells were transfected with expression plasmids pDEST-FLAG-pCR3 DNA2 (wild-type, C136S or C396S) as bait or GFP-pCR3 vector as control (5 µg) using calcium phosphate. After 24 h, cells were detached in ice-cold PBS and collected by centrifugation. Cell pellets were lysed in 100 µl of lysis buffer (100 mM NaCl, 50 mM Tris–HCl (pH 8.0), 10% glycerol, 1 mM EDTA, 0.1% NP-40, 2 mM DTT, cOmplete Mini Protease Inhibitor Cocktail tablets (Roche) and 0.1% Benzonase (Santa Cruz)) and incubated on ice for 30 min. The lysates were centrifuged at 17,200×g and 4 °C for 30 min. Immunoprecipitation was performed with 20 µl anti-FLAG M2 beads (Sigma Aldrich), rotating for 1 h at 4 °C. The beads were then washed six times with 1 ml of wash buffer (100 mM NaCl, 50 mM Tris–HCl (pH 8.0), 10% glycerol, 1 mM EDTA, 2 mM DTT). The immunoprecipitates were eluted in wash buffer supplemented with

200 ng/µl 3x FLAG peptide (Sigma) for 1 h at 4 °C. Whole cell extracts and immunoprecipitates were analysed by SDS–PAGE and Western blotting.

**YFP-DNA2-FLAG-DNA2 co-immunoprecipitation**. HEK293T cells were transfected with pDEST-YFP/FRT/TO DNA2 (WT/C396S) as bait using calcium phosphate. After 24 h, cells were collected and lysed as for the DNA2–CIA targeting complex co-immunoprecipitation experiment. The lysates were incubated with 10 µl GFP-Trap agarose resin (Chromotek) for 1 h at 4 °C rotating. The beads were then washed four times for 5 min with 1 ml of wash buffer (100 mM NaCl, 50 mM Tris–HCl (pH 8.0), 10% glycerol, 1 mM EDTA, 0.1% NP-40, 2 mM DTT). 50 µl of 25 nM purified FLAG-tagged wild-type DNA2 and DNA2 C396S, respectively, was added to the immunoprecipitated samples and incubated for 1 h at 4 °C. The beads were then washed four times for 5 min with 1 ml of wash buffer (as above). The immunoprecipitates were eluted in 40 µl SDS–PAGE sample buffer, and analysed by SDS–PAGE and Western blotting.

**Protein stability time course**. HeLa Flp-In T-REx cells inducibly expressing FLAG-DNA2 WT and FLAG-DNA2 C396S were treated with 1 µg/ml doxycycline to induce protein expression. After 16 h, 100 µg/ml cycloheximide was added, and cells were collected at different time points (0, 2, 4, 6, 8 h after cycloheximide addition). Protein levels were detected by SDS–PAGE and Western blotting.

**Western blots and antibodies**. Western blots were imaged on a FUSION SOLO (Witec AG) chemiluminescence imaging system. The following primary antibodies (at a dilution of 1:1000) were used: β-actin-horseradish peroxidase (HRP) (C4, sc-47778; Santa Cruz Biotechnology), DNA2 (ab96488; Abcam), CIAO1 (ab83088; Abcam), MIP18/FAM96b (20108-1-AP; Proteintech), MMS19 (16015-1-AP; Proteintech). Secondary antibodies (at a dilution of 1: 2500) were anti-Mouse-HRP (NA931; Amersham) and anti-Rabbit-HRP (NA934; Amersham). Uncropped blots can be found in Supplementary Fig. 6.

**Reporting summary**. Further information on research design is available in the Nature Research Reporting Summary linked to this article.

## Data availability

Source data are available as Supplementary Data 1. Full blots are shown in Supplementary Information. All other data are available from the corresponding authors upon reasonable request.

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

## Acknowledgements

We thank Susanne Kassube for the kind gift of *Hi5* insect cell pellets expressing *mDna2*, Stephen S. Taylor for HeLa Flp-In T-REx cells, and the Functional Genomics Center Zurich for mass spectrometry analyses. We are grateful to all members of the Gari lab for helpful discussions. This project has received funding from the Swiss National Science Foundation (PP00P3_144784/1 and PP00P3_172959/1), the Human Frontier Science Programme (CDA00043/2013-C), the Olga-Mayenfisch foundation, and the University of Zurich. L.M. is recipient of a Postdoc fellowship of the University of Zurich (FK-18-050). The lab of K.G. is part of COST action CA15133. Research in the lab of P.C. is supported by the Swiss National Science Foundation (31003A_175444) and a European Research Council Consolidator Grant (681630).

## Author contributions

L.M. and K.G. conceived the project and designed experiments. L.M., S.W., G.B., A.P., I.C., S.K., R.L. and K.G. performed experiments. L.M., S.W., G.B., A.P., I.C., P.C. and K.G. analysed and interpreted data. L.M. and K.G. prepared figures and wrote the manuscript. All authors read and edited the manuscript. K.G. supervised the study.

## Competing interests

The authors declare no competing interests.
