## [Peer Review File · Communications Biology]

Reviewers' comments:

Reviewer #1 (Remarks to the Author):

The nuclease activity of DNA2 is required for the effective repair of DNA damage during DSB repair. In addition, DNA2 is required for Okazaki fragment processing and the processing of stalled replication forks during DNA replication. How the ATPase and helicase functions of DNA2 are coordinated is not fully understood. Building upon previous structural work on mouse DNA2 (Zhou et al. 2015) and biochemical work with the related AddB nuclease (Yeeles et al. 2009), Mariotti et al. report a structural role for the FeS cluster in human DNA2. Loss of the FeS cluster in human DNA2 is shown to reduce its affinity for a variety of substrates with 5' and 3' ssDNA overhangs. Based on limited proteolysis and altered migration profiles in EMSA they propose that FeS cluster loss induces a conformational change in DNA2 that distorts the ssDNA binding channel. FeS cluster loss is also shown to impair ATPase and helicase activity. The authors go on to characterise the ability of DNA2 to interact with DNA in a reversibly redox sensitive fashion. They show that oxidation of Dna2 impairs DNA binding and that reduction restores binding activity. Interestingly, this occurs independently of the FeS cluster, however the mechanism is not explored further.

Overall the work highlights an important role of the human DNA2 FeS cluster in DNA binding, ATPase and helicase activity. The data also supports a structural role for the cluster. These are not particularly surprising findings given prior work with the yeast protein (Pokharel and Campbell, 2012) and the crystal structure of the mouse protein (Zhou et al. 2015). The data is generally of good quality and most of the conclusions are sound. However, the data relating the DNA binding preferences of the FeS mutants are not so compelling. Perhaps the most interesting observation is the redox sensitive ability of DNA2 to bind DNA, independently of the FeS cluster. Generally, this is a sound piece of work that will be of most interest to researchers within the specific field.

Specific comments

- The authors state: "The purified DNA2 proteins, including the FeS cluster-deficient variants, were however stable in SDS- and Native-PAGE analyses". Can the authors be confident that the cysteine mutants do not increase protein aggregation? It looks like there might be more material in the wells in the native gel in Figure S1D with the C396S mutant. The reduction in DNA binding affinity may be as a result of the production of misfolded protein. This is partially addressed by the Flag-DNA2, CIA pulldowns, however. The authors could include gel filtration traces for their cysteine mutants indicating that there is a negligible change in the size of the void peak relative to WT (as shown in supplementary figure 2A).
- The EMSA data related to altered DNA binding preferences for the mutants isn't very convincing. For example, the wild type protein binds better as the 5' overhang is increased, as does the mutant, so I'm not sure this says anything specific about the properties of the mutant. Similarly, binding of the wild type to the 3' overhang is greatly reduced compared to the 5' overhang. Although it is almost undetectable for the mutant on the 3' overhang, this is not really surprising as the specific product with the 5' overhang is very weak. Again, I'm not convinced this indicates the mutant has different properties to the wild type.
- The authors use limited proteolysis to analyse the cleavage of DNA2 WT and mutants. Figure 2D shows the presence of an additional band at approximately 90kDa in the C136S and C396S lanes prior to proteolysis at a similar molecular weight to the band that appears for WT DNA2 following digestion. What is this band? Is it a contaminant or a breakdown product? If it is a breakdown product it undermines the claim that the DNA2 cysteine mutants do not affect protein stability. The full gel for these experiments should be displayed in sup information.
- Based on limited proteolysis and migration in EMSA it is concluded that there is a conformational

change when the FeS cluster is lost. However, there is no difference in migration in native PAGE. Do the authors have a suggestion as to why this might be?

- It would be useful if the different DNA substrates used in EMSA could be shown in the relevant figures.
- Line 181 should read Figure 3A
- Line 189 needs an associated figure and reference

Reviewer #2 (Remarks to the Author):

This manuscript reports that loss of the FeS cluster in human DNA2 inactivates the enzyme with respect to nuclease, helicase and ATPase activity. These conclusions are drawn from biochemical studies using different assays.

The authors further conclude that the FeS cluster in human DNA2 causes a conformational change that impairs its ability to efficiently bind to DNA substrates. This is different from the situation in yeast DNA2.

The paper is well written, but I cannot follow the argumentation of the authors which is solely based on biochemical assay studies and chromatography. No X-ray structure data are presented to support a conformational change upon FeS cluster loss. This might be difficult at the present stage, but there is also no spectroscopic evidence presented. I am also not convinced that the role of the 4Fe4S cluster is purely structural, since its redox properties were not investigated by spectroscopy (e.g. UV-Vis, EPR, X-ray absorption, Mössbauer, or NMR spectroscopy).

Therefore, I cannot recommend publication in Communications Biology.

Your manuscript entitled "A structural, non-redox role for the iron-sulphur cluster in human DNA2" has now been seen by 2 referees, whose comments are appended below. You will see from their comments copied below that while they find your work of considerable potential interest, they have raised quite substantial concerns that must be addressed. In light of these comments, we cannot accept the manuscript for publication, but would be interested in considering a revised version that addresses these concerns.

We hope you will find the referees' comments useful as you decide how to proceed. We would be happy to look at a substantially revised manuscript that addresses the main issues raised by reviewer 2. Please provide spectroscopic evidence for the presence of the iron-sulphur cluster. We also encourage you to investigate the redox properties by spectroscopy (e.g. UV-Vis, EPR, X-ray absorption, Mössbauer, or NMR spectroscopy) to support your hypothesis that the role of the FeS cluster in DNA2 is purely structural. Please bear in mind that we will be reluctant to approach the referees again in the absence of major revisions. If the revision process takes significantly longer than six months, we will be happy to reconsider your paper at a later date, as long as nothing similar has been accepted for publication at Communications Biology or published elsewhere in the meantime.

Reviewers' comments:

Reviewer #1 (Remarks to the Author):

The nuclease activity of DNA2 is required for the effective repair of DNA damage during DSB repair. In addition, DNA2 is required for Okazaki fragment processing and the processing of stalled replication forks during DNA replication. How the ATPase and helicase functions of DNA2 are coordinated is not fully understood. Building upon previous structural work on mouse DNA2 (Zhou et al. 2015) and biochemical work with the related AddB nuclease (Yeeles et al. 2009), Mariotti et al. report a structural role for the FeS cluster in human DNA2. Loss of the FeS cluster in human DNA2 is shown to reduce its affinity for a variety of substrates with 5' and 3' ssDNA overhangs. Based on limited proteolysis and altered migration profiles in EMSA they propose that FeS cluster loss induces a conformational change in DNA2 that distorts the ssDNA binding channel. FeS cluster loss is also shown to impair ATPase and helicase activity. The authors go on to characterise the ability of DNA2 to interact with DNA in a reversibly redox sensitive fashion. They show that oxidation of Dna2 impairs DNA binding and that reduction restores binding activity. Interestingly, this occurs independently of the FeS cluster, however the mechanism is not explored further.

Overall the work highlights an important role of the human DNA2 FeS cluster in DNA binding, ATPase and helicase activity. The data also supports a structural role for the cluster. These are not particularly surprising findings given prior work with the yeast protein (Pokharel and Campbell, 2012) and the crystal structure of the mouse protein (Zhou et al. 2015). The data is generally of good quality and most of the conclusions are sound. However, the data relating the DNA binding preferences of the FeS mutants are not so compelling. Perhaps the most interesting observation is the redox sensitive ability of DNA2 to bind DNA, independently of the FeS cluster. Generally, this is a sound piece of work that will be of most interest to researchers within the specific field.

We thank the reviewer for their positive assessment. We agree that the observed redox sensitivity is very interesting and have added some additional data (Fig. 5b) showing that DNA2 is much less sensitive to oxidation when bound to DNA first, suggesting that – once DNA2 is in a DNA-bound conformation – any redox-sensitive residue(s) are locked in place or protected from oxidation. In light of these results, we then wondered whether there was a cysteine residue located close to the DNA binding tunnel, which could act as a redox sensor, and investigated the most likely candidate cysteine-743 further (Supplementary Fig. 5). DNA binding by the C743S variant was however still sensitive to oxidation, rendering a redox-sensor function by cysteine-743 unlikely. Although a negative result, we thought to add this piece of information as a supplementary figure to the manuscript, since it would help direct the research efforts of others interested in following up on DNA2's oxidation sensitivity.

Specific comments

- The authors state: "The purified DNA2 proteins, including the FeS cluster-deficient variants, were however stable in SDS- and Native-PAGE analyses". Can the authors be confident that the cysteine mutants do not increase protein aggregation? It looks like there might be more material in the wells in the native gel in Figure S1D with the

C396S mutant. The reduction in DNA binding affinity may be as a result of the production of misfolded protein. This is partially addressed by the Flag-DNA2, CIA pulldowns, however. The authors could include gel filtration traces for their cysteine mutants indicating that there is a negligible change in the size of the void peak relative to WT (as shown in supplementary figure 2A).

We agree with the reviewer that it is important to distinguish whether the FeS cluster-deficient variants bind DNA less well because they have a reduced affinity or because a major part of the purified protein is aggregated.

It should be noted that in all EMSAs in the manuscript, we have used proteins eluted from FLAG beads (without any further purification or concentration steps) – to limit in time the purification protocol and to maintain intact FeS clusters (see Supplementary Fig. 1c for protein purity). To check whether the C396S variant tends to aggregate more, we loaded thus purified WT DNA2 and DNA2 C396S on a S200 gel filtration column (Supplementary Fig. 2a). While the C396S variant – surprisingly – eluted at a higher estimated molecular weight than the wild-type protein, there was no obvious difference in the amount of protein that eluted in the void fractions, as visualised by SDS-PAGE and InstantBlue staining (unfortunately, in this setup, the UV signal did not match the elution profile, possibly because the protein amounts used were very small, although we cannot exclude a general problem with the UV detector/lamp). We can, hence, exclude an increase in protein aggregation upon FeS cluster loss.

This notion is also supported by our new Native-PAGE experiment (see below), in which the samples were simply ran for longer and no difference in the amount of material found in the wells was discernible between wild-type DNA2 and the C396S variant (see also answer to point 4).

Of note, the gel filtration profile of wild-type DNA2 (Supplementary Fig. 2a) looks different from what we had shown in the original version where we had a better ratio of DNA2 (main peak) over DNA2 (void fractions). The main difference is that in the original experiment, we wanted to see whether WT DNA2 could oligomerise and had therefore concentrated and spun down the protein (purified from a much larger starting culture) prior to injection on the gel filtration column, whereas in the new experiment we wanted to reflect most closely the aggregation state of the proteins used in EMSAs and have therefore omitted the concentration and centrifugation steps prior to injection.

We have also amended the manuscript text to address a potential protein aggregation problem:

Page 3:

The approximately 10-fold reduction in DNA binding that we observed with the FeS cluster-deficient DNA2 variants may indicate that the affinity for DNA is greatly reduced in the absence of an FeS cluster and that DNA binding is only favoured at high protein concentrations. However, to rule out the possibility that only a subpopulation of the FeS cluster-deficient DNA2 variants is properly folded and, hence, proficient in DNA binding, whereas the majority is misfolded and tends to aggregate, we analysed both wild-type DNA2 and the C396S variant by size exclusion chromatography (Supplementary Fig. 2a). Although in both cases some protein was found in the void peak, which

is indicative of protein aggregation, the majority of wild-type DNA2 and the C396S variant eluted as a single peak after the void fractions. Surprisingly, however, the C396S variant eluted at a higher estimated molecular weight from the gel filtration column than the wild-type protein.

- The EMSA data related to altered DNA binding preferences for the mutants isn't very convincing. For example, the wild type protein binds better as the 5' overhang is increased, as does the mutant, so I'm not sure this says anything specific about the properties of the mutant. Similarly, binding of the wild type to the 3' overhang is greatly reduced compared to the 5' overhang. Although it is almost undetectable for the mutant on the 3' overhang, this is not really surprising as the specific product with the 5' overhang is very weak. Again, I'm not convinced this indicates the mutant has different properties to the wild type.

We agree with the reviewer that the experiments in the original manuscript were not the best way of illustrating a difference between wild-type DNA and FeS cluster-deficient DNA2. In the new version, we have removed these data and instead include EMSAs using 5' overhang DNA substrates with a wider range of ssDNA portions (10 to 40 nucleotides, instead of 20 to 30 nucleotides). These new data (Fig. 3a-c) show very clearly a largely reduced ability of FeS cluster-deficient DNA2 to bind to shorter DNA substrates.

Page 4:

We therefore used 5' overhang DNA substrates with single-stranded portions of 10 to 40 nucleotides. To achieve comparable DNA binding between wild-type DNA2 and DNA2 C396S, we used a 10-fold higher protein concentration for DNA2 C396S. We observed partial DNA binding by wild-type DNA2 for substrates with a minimal overhang of 15 nucleotides and complete binding for substrates with overhangs of 20 and more nucleotides (Fig. 3a,c). In contrast, the C396S variant displayed little binding to substrates with 15 or 20 nucleotides overhangs, and complete binding could only be observed with overhangs of more than 35 nucleotides (Fig. 3b,c). Taken together, these findings suggest that FeS cluster-deficient DNA2 binds less efficiently to shorter DNA substrates, likely due to a conformational change affecting the DNA binding tunnel.

- The authors use limited proteolysis to analyse the cleavage of DNA2 WT and mutants. Figure 2D shows the presence of an additional band at approximately 90kDa in the C136S and C396S lanes prior to proteolysis at a similar molecular weight to the band that appears for WT DNA2 following digestion. What is this band? Is it a contaminant or a breakdown product? If it is a breakdown product it undermines the claim that the DNA2 cysteine mutants do not affect protein stability. The full gel for the these experiments should be displayed in sup information.

We analysed the band at approximately 90 kDa (from the C136S sample prior to limited proteolysis) by mass spectrometry – it corresponds to HSP83. We indicate this in the figure legend (Fig. 2d).

We now also include a line scan profile to illustrate better that the proteolytic product is unique to DNA2 WT (Fig. 2d) and show the full gel of the experiment (Supplementary Fig. 2d).

- Based on limited proteolysis and migration in EMSA it is concluded that there is a conformational change when the FeS cluster is lost. However, there is no difference in migration in native PAGE. Do the authors have a suggestion as to why this might be?

Initially, we had done the Native-PAGE experiment to check that the proteins were not degraded. We thought that the fact that both WT and C396S migrated not very far in the gel and to similar levels was due to the isoelectric point of 7.54. In light of our new gel filtration results, we ran the native PAGE for twice as long to see if WT and C396S would migrate differently, which was however not the case (see also answer to point 1).

We have added a tentative explanation to the manuscript:

Page 4:

In line with a conformational change upon FeS cluster loss, the protein/DNA complexes formed by wild-type DNA2 and the DNA2 variants, respectively, differed in their migration patterns (Fig. 2a,b,c). Protein/DNA complexes formed by the FeS cluster-deficient variants migrated further in the gel (Fig. 2a,b,c). Although it may seem counter-

intuitive at first sight given that the C396S variant eluted at a higher estimated molecular weight (Supplementary Fig. 2a), elution patterns in size exclusion chromatography depend on the molecular weight and the shape of the protein, while in native gel EMSAs the exposed charge of the protein plays an additional role. It is hence possible that the exposed charge of DNA2 changes upon FeS cluster loss, which leads to the increased migration observed in EMSAs (Fig. 2a,b,c). In line with this idea, in Native-PAGE where protein size, shape and charge play a role, wild-type DNA2 and the C396S variant migrate very similarly (Supplementary Fig. 2c), raising the possibility that the presumably more elongated shape of the C396S variant, which should result in a more retarded migration in Native-PAGE, may be counterbalanced by an exposed charge that increases its mobility.

- It would be useful if the different DNA substrates used in EMSA could be shown in the relevant figures.

We agree and have added the DNA substrates.

- Line 181 should read Figure 3A

We have changed this.

- Line 189 needs an associated figure and reference

We have added the figure information.

Reviewer #2 (Remarks to the Author):

This manuscript reports that loss of the FeS cluster in human DNA2 inactivates the enzyme with respect to nuclease, helicase and ATPase activity. These conclusions are drawn from biochemical studies using different assays.

The authors further conclude that the FeS cluster in human DNA2 causes a conformational change that impairs its ability to efficiently bind to DNA substrates. This is different from the situation in yeast DNA2.

The paper is well written, but I cannot follow the argumentation of the authors which is solely based on biochemical assay studies and chromatography. No X-ray structure data are presented to support a conformational change upon FeS cluster loss. This might be difficult at the present stage, but there is also no spectroscopic evidence presented. I am also not convinced that the role of the 4Fe4S cluster is purely structural, since its redox properties were not investigated by spectroscopy (e.g. UV-Vis, EPR, X-ray absorption, Mössbauer, or NMR spectroscopy). Therefore, I cannot recommend publication in *Communications Biology*.

We agree with the reviewer that the original title was perhaps a bit bold in that it excluded any potential redox role that the FeS cluster in DNA2 could have and that we might have missed with our experimental approach. The new title of the manuscript ("Iron-sulphur cluster binding induces a conformational change in human DNA2") reflects the content of the manuscript better since it puts the emphasis on the conformational change that we observe upon FeS cluster loss/binding and that we have been able to substantiate by additional data (see also detailed response to Reviewer 1).

Briefly, our new data show:

- a different elution pattern from an S200 gel filtration column when comparing WT DNA2 and the C396S variant (Supplementary Fig. 2a)
- a clear difference in ssDNA length requirement for DNA binding, with WT DNA2 displaying complete binding for substrates with overhangs of 20 and more nucleotides (Fig. 3a,c), whereas the C396S variant displayed complete binding only with overhangs of more than 35 nucleotides (Fig. 3b,c)

Unfortunately, however, the reviewer is right in their assessment that a more detailed structural analysis of a conformational change is hampered by the relatively low amount and concentration of purified human DNA2 that can be obtained. Even the most concentrated protein preparation was barely concentrated enough to visualise the presence of an FeS cluster in UV-vis spectroscopy:

However, we managed to obtain mouse Dna2 in a sufficiently high amount and concentration (Supplementary Fig. 4d) to address spectroscopically whether DNA2's FeS cluster is oxidation-sensitive. First, we confirmed that purified mDna2 binds to DNA with a similar affinity than human DNA2 (Supplementary Fig. 4e). We then showed that mDna2 loses its ability to bind DNA when incubated for 20 h at atmospheric oxygen (Fig. 5e), which could be fully recovered upon treatment with TCEP (Fig. 5e).

In UV-vis spectroscopy, we could then show that mDna2 displays a typical shoulder at 410 nm, indicative of the presence of an FeS cluster (Fig. 5f). Importantly, the spectra did not change when mDna2 was left to oxidise for 20 h or when re-reduced by TCEP after oxidation (Fig. 5f), confirming that the FeS cluster remains intact after 20 h of air exposure and that the observed oxidation sensitivity of DNA2/mDna2 is not mediated by the FeS cluster.

Reviewers' comments:

Reviewer #1 (Remarks to the Author):

General comments

We stated that “perhaps the most interesting observation is the redox sensitive ability of DNA2 to bind DNA, independently of the FeS cluster.”

In an attempt to further describe this redox sensitivity a single point mutation (C743S) was made in human DNA2 based upon its proximity to the DNA binding tunnel in the mouse DNA2 crystal structure. This mutant was still sensitive to oxidation. The authors did not design any further mutants. We maintain that this sensitivity is particularly interesting and that more efforts could be made by the authors to investigate this. However, considering that the focus of the paper concerns a conformational effect of iron-sulphur cluster loss, coupled with the fact that in the original review we did not request any specific experiments to investigate the FeS independent redox function, we are satisfied with these additional experiments.

Specific comments

We asked “Can the authors be confident that the cysteine mutants do not increase protein aggregation?”

In response the authors loaded purified WT DNA2 and DNA2 C396S on a S200 gel filtration column (Supplementary Fig. 2a) in order to compare the amount of protein in the void volumes. Unfortunately, the authors state that “the UV signal did not match the elution profile” and as such, they have omitted the traces and included only the gels of the column fractions. Because the gel bands are faint it is difficult to interpret the relative amounts of protein in the void between the WT and mutant. In order to increase the signal to noise, the authors could either silver stain the gels or repeat the experiment using a smaller bed volume column. Despite these criticisms, the new native-PAGE data does suggest that there is no difference in aggregation between the WT and mutant. In addition, because the bands are weak it is difficult to identify clear peaks. The authors state that “the C396S variant – surprisingly – eluted at a higher estimated molecular weight than the wild-type protein”. If the gel filtration were repeated and a trace/gel could be provided that clearly illustrates a change in elution volume between the WT and mutant, this would provide compelling evidence for a conformational change induced by FeS cluster loss that would strengthen the central argument of the paper. The authors note that in the original manuscript they included traces where the ratio of main to void peak for WT DNA2 was better than that of the new data discussed here. We appreciate their efforts to use unconcentrated material to reflect the aggregation state of the samples used in subsequent DNA binding experiments, however. We suggest that the authors might repeat the gel filtration with concentrated mutant DNA2 to obtain sufficient signal to unambiguously describe the change in elution volume for the soluble species that is suggested by the new gel filtration experiment.

We stated that “The EMSA data related to altered DNA binding preferences for the mutants isn't very convincing”

In response the authors “include EMSAs using 5' overhang DNA substrates with a wider range of ssDNA portions (10 to 40 nucleotides, instead of 20 to 30 nucleotides).” Authors used 200 nM C296S mutant and 20 nM WT for the 5' overhang binding experiments in figure 3 “to achieve comparable DNA binding between wild-type DNA2 and DNA2 C396S”. These values are presumably obtained from the data in figure 2 using a 5'-flap, yet the substrate used for the assays in figure 3 is a 5' overhang for which there is no binding titration provided. Without such a titration for the 5' overhang substrate the data is difficult to fully interpret. We suggest that these assays be repeated using the lowest

protein concentration for which 100% binding is observed for the longest (40 nt) overhang for both WT and mutant. This would aid comparison of the substrate preference of each protein at the various overhang lengths.

Identification of the 90 kDa band from the C136S sample prior to limited proteolysis as HSP83 is helpful and strengthens the argument that there is a difference in the gels of proteinase k treated mutants vs WT.

We appreciate the authors repeating the native PAGE with WT and mutant proteins to determine whether they migrated differently. Their explanation in the manuscript is sufficient justification for the lack of observed differences.

The change of the title to state that 'FeS cluster binding induces a conformational change...' is perhaps a bit misleading as the FeS cluster is an essential component of DNA2 as is presumably never lacking. We suggest that the authors adapt the title to refer to the FeS cluster as a crucial structural element.

The argument that FeS cluster loss induces a conformational change within DNA2 is consistent with previous work on AddB and mouse DNA2. The AddB and DNA2 nuclease structures are very similar and the FeS clusters adopt the same position, pinning the nuclease domain in both. The data presented here confirm that the FeS cluster has a structural role in human DNA2. They do not exclude an additional important redox function for the FeS cluster. If the suggested experiments were completed and gave results consistent with the arguments presented in the manuscript, the paper would be appropriate for publication.

Reviewer #2 (Remarks to the Author):

The authors have addressed my concerns now by presenting spectroscopic data. I follow the argumentation that the Protein cannot be prepared in quantities needed for advanced spectroscopic methods. However, I am happy to see that the authors use the mouse protein in order to Support their conclusion that the FeS Cluster is apparently not redox active by presenting UV-Vis spectroscopic data. I am also satisfied with all changes performed in the revised manuscript and suggest to publish the paper as it is.

Reviewers' comments:

Reviewer #1 (Remarks to the Author):

General comments

We stated that “perhaps the most interesting observation is the redox sensitive ability of DNA2 to bind DNA, independently of the FeS cluster.”

In an attempt to further describe this redox sensitivity a single point mutation (C743S) was made in human DNA2 based upon its proximity to the DNA binding tunnel in the mouse DNA2 crystal structure. This mutant was still sensitive to oxidation. The authors did not design any further mutants. We maintain that this sensitivity is particularly interesting and that more efforts could be made by the authors to investigate this. However, considering that the focus of the paper concerns a conformational effect of iron-sulphur cluster loss, coupled with the fact that in the original review we did not request any specific experiments to investigate the FeS independent redox function, we are satisfied with these additional experiments.

Specific comments

1. We asked “Can the authors be confident that the cysteine mutants do not increase protein aggregation?”

In response the authors loaded purified WT DNA2 and DNA2 C396S on a S200 gel filtration column (Supplementary Fig. 2a) in order to compare the amount of protein in the void volumes. Unfortunately, the authors state that “the UV signal did not match the elution profile” and as such, they have omitted the traces and included only the gels of the column fractions. Because the gel bands are faint it is difficult to interpret the relative amounts of protein in the void between the WT and mutant. In order to increase the signal to noise, the authors could either silver stain the gels or repeat the experiment using a smaller bed volume column. Despite these criticisms, the new native-PAGE data does suggest that there is no difference in aggregation between the WT and mutant.

In addition, because the bands are weak it is difficult to identify clear peaks. The authors state that “the C396S variant – surprisingly – eluted at a higher estimated molecular weight than the wild- type protein”. If the gel filtration were repeated and a trace/gel could be provided that clearly illustrates a change in elution volume between the WT and mutant, this would provide compelling evidence for a conformational change induced by FeS cluster loss that would strengthen the central argument of the paper. The authors note that in the original manuscript they included traces where the ratio of main to void peak for WT DNA2 was better than that of the new data discussed here. We appreciate their efforts to use unconcentrated material to reflect the aggregation state of the samples used in subsequent DNA binding experiments, however. We suggest that the authors might repeat the gel filtration with concentrated mutant DNA2 to obtain sufficient signal to unambiguously describe the change in elution volume for the soluble species that is suggested by the new gel filtration experiment.

We agree with the reviewer that the difference in elution pattern was not optimally visible in the gels shown in the original Supplementary Fig. 2a. We have now replaced the gels with better stained ones. In addition, we have also included a graph displaying the band intensities of the elution fractions. While such a representation is a bit unconventional, we think that it may help –together with the new gels– to visualise the different elution patterns.

Figure S2

2. We stated that “The EMSA data related to altered DNA binding preferences for the mutants isn’t very convincing”

In response the authors “include EMSAs using 5’ overhang DNA substrates with a wider range of ssDNA portions (10 to 40 nucleotides, instead of 20 to 30 nucleotides).” Authors used 200 nM C296S mutant and 20 nM WT for the 5’ overhang binding experiments in figure 3 “to achieve comparable DNA binding between wild-type DNA2 and DNA2 C396S”. These values are presumably obtained from the data in figure 2 using a 5’-flap, yet the substrate used for the assays in figure 3 is a 5’ overhang for which there is no binding titration provided. Without such a titration for the 5’ overhang substrate the data is difficult to fully interpret. We suggest that these assays be repeated using the lowest

protein concentration for which 100% binding is observed for the longest (40 nt) overhang for both WT and mutant. This would aid comparison of the substrate preference of each protein at the various overhang lengths.

The crystal structure of mouse Dna2 with ssDNA shows that Dna2 cannot bind dsDNA because its DNA binding tunnel is too narrow (Zhou et al., eLife, 2015). Moreover, the structure/function analysis of Dna2 suggests that a 5' ssDNA overhang is threaded through the DNA binding tunnel formed by the nuclease domain until it reaches the DNA binding site of the helicase domain where the DNA-protein interaction is stabilised. It would hence appear that the stability of DNA binding is primarily dependent on the length of the single-stranded DNA portion, with a certain minimal length required for the ssDNA to reach the helicase domain.

Based on the results of this study and given the high conservation between the mouse and human protein (80% identity), we assume that DNA2 binds a 5' overhang substrate the same as a 5' flap substrate, since it is the 5' single-stranded portion of the substrate that is engaged in the DNA binding tunnel and there does not seem to be any contribution from dsDNA to the DNA-protein interaction. We used 5' overhang substrates of different ssDNA lengths instead of various flap substrates simply because it required the synthesis of less and shorter oligos, while the important part (the 5' ssDNA overhang) was the same. To exclude any possible effects of sequence composition to DNA binding (for example through the formation of secondary structures), we notably made sure that the ssDNA portions in the overhang substrates (starting from the 5' end of the overhang) were identical to the sequence in the flap substrate.

Having said this, under normal circumstances, it would require very little effort to perform titration EMSAs with 5' overhang substrates and we would happily include them as a supplementary figure. Given the current situation, however, including them would mean to delay publication of the manuscript considerably. We would thus hope that the manuscript is acceptable without them.

3. Identification of the 90 kDa band from the C136S sample prior to limited proteolysis as HSP83 is helpful and strengthens the argument that there is a difference in the gels of proteinase k treated mutants vs WT.

4. We appreciate the authors repeating the native PAGE with WT and mutant proteins to determine whether they migrated differently. Their explanation in the manuscript is sufficient justification for the lack of observed differences.

5. The change of the title to state that 'FeS cluster binding induces a conformational change...' is perhaps a bit misleading as the FeS cluster is an essential component of DNA2 as is presumably never lacking. We suggest that the authors adapt the title to refer to the FeS cluster as a crucial structural element.

We understand the reviewer's concern and would suggest to change the title to "The iron-sulfur cluster in human DNA2 is an important structural element required for all biochemical activities".

6. The argument that FeS cluster loss induces a conformational change within DNA2 is consistent with previous work on AddB and mouse DNA2. The AddB and DNA2 nuclease structures are very similar and the FeS clusters adopt the same position, pinning the nuclease domain in both. The data presented here confirm that the FeS cluster has a structural role in human DNA2. They do not exclude an additional important redox function for the FeS cluster. If the suggested experiments were completed and gave results consistent with the arguments presented in the manuscript, the paper would be appropriate for publication.

Reviewer #2 (Remarks to the Author):

The authors have addressed my concerns now by presenting spectroscopic data. I follow the argumentation that the Protein cannot be prepared in quantities needed for advanced spectroscopic methods. However, I am happy to see that the authors use the mouse protein in order to Support their conclusion that the FeS Cluster is apparently not redox active by presenting UV-Vis spectroscopic data. I am also satisfied with all changes performed in the revised manuscript and suggest to publish the paper as it is.

We thank both reviewers for the time they have taken to evaluate and improve the manuscript.